# Adenosine A_2A_ Receptors Are Upregulated in Peripheral Blood Mononuclear Cells from Atrial Fibrillation Patients

**DOI:** 10.3390/ijms22073467

**Published:** 2021-03-27

**Authors:** Héctor Godoy-Marín, Romain Duroux, Kenneth A. Jacobson, Concepció Soler, Hildegard Colino-Lage, Veronica Jiménez-Sábado, José Montiel, Leif Hove-Madsen, Francisco Ciruela

**Affiliations:** 1Pharmacology Unit, Department of Pathology and Experimental Therapeutics, Faculty of Medicine and Health Sciences, Institute of Neurosciences, University of Barcelona, 08907 L’Hospitalet de Llobregat, Spain; hectorgodoymarin13@gmail.com; 2Neuropharmacology & Pain Group, Neuroscience Program, Institut d’Investigació Biomèdica de Bellvitge, IDIBELL, 08907 L’Hospitalet de Llobregat, Spain; concepciosoler@ub.edu; 3Molecular Recognition Section, Laboratory of Bioorganic Chemistry, National Institute of Diabetes and Digestive and Kidney Diseases, National Institutes of Health, Bethesda, MD 20892, USA; romain.duroux@gmail.com (R.D.); kennethj@niddk.nih.gov (K.A.J.); 4Immunology Unit, Department of Pathology and Experimental Therapeutics, Faculty of Medicine and Health Sciences, Institute of Neurosciences, University of Barcelona, 08907 L’Hospitalet de Llobregat, Spain; 5Barcelona Biomedical Research Institute, IIBB-CSIC, 08036 Barcelona, Spain; HColino@santpau.cat; 6Biomedical Research Institute Sant Pau, IIB Sant Pau, 08025 Barcelona, Spain; VJimenezS@santpau.cat; 7CIBERCV, 28029 Madrid, Spain; 8Department Cardiac Surgery, Hospital de la Santa Creu i Sant Pau, 08036 Barcelona, Spain; JMontiel@santpau.cat

**Keywords:** adenosine A2A receptor, atrial fibrillation, adenosine deaminase, peripheral blood mononuclear cells, adenosine

## Abstract

Atrial fibrillation (AF) is the most common form of cardiac arrhythmia seen in clinical practice. While some clinical parameters may predict the transition from paroxysmal to persistent AF, the molecular mechanisms behind the AF perpetuation are poorly understood. Thus, oxidative stress, calcium overload and inflammation, among others, are believed to be involved in AF-induced atrial remodelling. Interestingly, adenosine and its receptors have also been related to AF development and perpetuation. Here, we investigated the expression of adenosine A2A receptor (A2AR) both in right atrium biopsies and peripheral blood mononuclear cells (PBMCs) from non-dilated sinus rhythm (ndSR), dilated sinus rhythm (dSR) and AF patients. In addition, plasma adenosine content and adenosine deaminase (ADA) activity in these subjects were also determined. Our results revealed increased A2AR expression in the right atrium from AF patients, as previously described. Interestingly, increased levels of adenosine content and reduced ADA activity in plasma from AF patients were detected. An increase was observed when A2AR expression was assessed in PBMCs from AF subjects. Importantly, a positive correlation (*p* = 0.001) between A2AR expression in the right atrium and PBMCs was observed. Overall, these results highlight the importance of the A2AR in AF and suggest that the evaluation of this receptor in PBMCs may be potentially be useful in monitoring disease severity and the efficacy of pharmacological treatments in AF patients.

## 1. Introduction

Atrial fibrillation (AF) is one of the most common cardiac arrhythmias, which is currently thought to affect approximately 2% of the world population. Its prevalence increases with age, affecting around 0.14% of people under 49 years old and increasing to 10-17% in people aged 80 or older [1]. This arrhythmia is known to cause an irregular and often abnormal fast heart rate and increases the risk of stroke, heart failure and other heart-related complications. The mechanisms underlying AF are complex and partially unresolved, but they involve electrical and structural remodelling (fibrosis) [2] that chiefly provokes a desynchronized activation of the atrium due to a shortening of the action potential wavelength, thus, facilitating fast and irregular beating [3]. AF episodes typically occur when arrhythmogenic substrates, such as fibrosis and inflammation coincide [4]. Indeed, inflammation has been associated with AF-pathological processes, as well as oxidative stress or thrombogenesis [5]. Therefore, it has been shown that the prevalence and prognosis of AF are highly associated with plasma levels of inflammatory biomarkers. Interestingly, as in any other inflammatory event, lymphocytes are recruited to the heart environment and play a key role in both stimulating and inhibiting inflammatory responses during the initiation, maintenance, and progression of AF. Therefore, the neutrophils to lymphocyte ratio becomes an actual outcome predictor in AF [6]. Notably, lymphocytes are also affected by the atrium inflammatory environment, which courses with an increased adenosine production and release, thus, inducing plastic changes in the lymphocytes during this process [7].

Adenosine is a purine nucleoside generated mostly by ATP catabolism and regulated by adenosine deaminase (ADA), adenosine kinase (ADK) and S-adenosyl-l-homocysteine hydrolase (SAHase) [8]. Adenosine levels are dramatically increased during inflammation [9,10], hypoxia [11], ischemia [12] and β-adrenergic receptor stimulation [13], and its release into the heart induces profound alterations in cardiomyocyte physiology, which may eventually evolve into AF development [14,15]. Since extracellular actions of adenosine are exerted through G protein-coupled adenosine receptors (ARs), namely A_1_R, A_2A_R, A_2B_R and A_3_R, some of these have been related to AF physiopathology. For instance, it has been demonstrated that A_2A_R density is increased in AF patients, leading to an abnormal calcium handling in the cardiomyocyte and triggering heart disfunctions [16,17]. Therefore, it is well-accepted that A_2A_R overexpression and activation by selective agonists may favour the occurrence of cardiac arrhythmias [18]. Conversely, the activation of A_1_R by intrinsic adenosine modifies atrial electrophysiology and promotes AF [19]. Accordingly, it has been postulated that adenosine shows a Janus-faced cardiac behaviour: Protection against ischemic damage but induction of cardiac arrhythmias [20].

Here, we set out to assess the status of the adenosinergic system in AF by analysing the expression of A_2A_Rs both in right atrium biopsies and peripheral blood mononuclear cells (PBMCs) from AF patients. In addition, the adenosine levels and adenosine deaminase (ADA) activity in plasma from the same patients were evaluated.

## 2. Results

### Adenosinergic System in AF

We previously demonstrated that AF patients showed an increased expression of A_2A_R in right atrial tissue [17], thus, suggesting a participation of this receptor in the development of AF. Now, we aimed at investigating a potential cardiac-peripheric connection of the adenosinergic system in AF. First, we confirmed a significant increase of A_2A_R density (*p* < 0.0001) in the right atrium from AF patients when compared to non-AF subjects (i.e., non-dilated sinus rhythm, ndSR) (Figure 1A,B) (see Table 1 for cardiac diseases related patients information). Indeed, a concomitant significant (*p* = 0.0207) increase in A_2A_R mRNA expression in right atrial tissue from these patients was also found (Figure 1C), consistent with the immunoblot results (Figure 1).

Next, we interrogated whether adenosine homeostasis was also altered in these patients. To this end, ADA activity and adenosine content in plasma from patients with, and without, AF were determined. Interestingly, a significant 38% reduction in ADA activity was found in plasma from AF patients when compared to ndSR subjects (*p* = 0.0286, F_(2, 66)_ = 3.753) (Figure 2A). Yet, no significant alteration in ADA activity was found in patients displaying atrial dilation (i.e., dilated sinus rhythm, dSR) (*p* = 0.1312) but not AF (Figure 2A). Subsequently, when the plasma adenosine content was evaluated, the levels in patients with AF were ~2.5-fold higher when compared to ndSR and dSR subjects (*p* < 0.0001, F_(2, 40)_ = 44.66) (Figure 2B). Indeed, a negative correlation (r = −0.4087, *p* = 0.0202) between ADA activity and adenosine content within ndSR and AF patients was found (Figure 2C). Collectively, these results suggested that concomitantly to an increased A_2A_R expression in right atrium, AF patients also show reduced ADA activity and increased adenosine content in plasma. 

Once the adenosine content and ADA activity in plasma from AF subjects was altered, we next questioned whether this dysregulation may eventually have any impact on A_2A_R density in PBMCs from these patients. To this end, we used a fluorescent A_2A_R antagonist, namely MRS7396 [21], to monitor A_2A_R density in PBMCs. First, we validated the specificity of MRS7396 A_2A_R staining in HEK293-A_2A_R^SNAP^ cells by displacing its binding with ZM241385, a selective A_2A_R antagonist (Figure 3). Interestingly, when PBMCs from a healthy subject were stained with MRS7396 we observed similar results (Figure 3). Therefore, the antagonist ZM241385 was able to displace the binding of the fluorescent A_2A_R ligand.

Subsequently, we stained PBMCs from ndSR, dSR and AF patients with MRS7396. PBMCs were further gated by flow cytometry based on morphology and the MRS7396 staining was determined in the absence or presence ZM241385 to define the A_2A_R specific binding in PBMCs from ndSR, dSR and AF patients (Figure 4A). The A_2A_R density in PBMCs from AF patients was significantly increased (~3.7-fold) when compared to both ndSR and dSR patients (*p* < 0.0001, F_(2, 39)_ = 23.55) (Figure 4A). Next, we assessed A_2A_R mRNA expression in PBMCs from ndSR, dSR and AF patients through RT-qPCR analysis A three-fold significant increase in A_2A_R mRNA expression in PBMCs from AF patients was found (*p* < 0.0001, F_(2, 24)_ = 14.64) (Figure 4B).

Importantly, a positive correlation between A_2A_R density in PBMCs and in right atrium of the same patients was observed (r = 0.7813, *p* = 0.001) (Figure 5A), thus indicating that changes of peripheral A_2A_R density may mirror changes occurring in the diseased atrium. In addition, a positive correlation between plasma adenosine content and A_2A_R density in PBMCs and in right atrium (r = 0.7650, *p* < 0.0001 and r = 0.7922, *p* = 0.0007, respectively) was found (Figure 5B,C). Finally, no correlation between plasma ADA activity and A_2A_R density in PBMCs and in right atrium was observed (Figure 5D,E). Overall, these results suggested a relationship between atrial and PBMCs A_2A_R expression, which might be related to adenosine content. 

## 3. Discussion

While cardiac A_2A_Rs play a key role in coronary flow and force generation, they have been also associated with some cardiopathological conditions, such as the genesis of arrhythmias [17,22]. Here, we assessed the status of the peripheric adenosinergic system in the most common type of cardiac arrhythmia seen in clinical practice, namely the AF. We hypothesized that the cardiac and peripherial adenosinergic systems may interact in AF. Indeed, immunoblot and RT-qPCR analysis of cardiac A_2A_R revealed once again an increased receptor density and RNA expression in the in right atrium biopsies from AF patients, in agreement with previous work [16]. Interestingly, the expression of A_2A_R in PMBCs from AF patients was increased when compared to ndSR and dSR subjects, thus, mirroring that observed in right atrium. In addition, the adenosine levels in plasma from AF patients were also increased and correlated with A_2A_R density in PBMCs and in right atrium. Collectively, these results point towards the existence of a potential cardiac-peripheral connection with the adenosinergic system in both health and disease and highlight the importance of the A_2A_R in AF.

Importantly, signalling through A_2A_R regulates PBMCs function [23], and increased levels of A_2A_R have been found in lymphocytes from patients with immunological diseases, including systemic lupus erythematosus [24], amyotrophic lateral sclerosis [25], multiple sclerosis [26] and rheumatoid arthritis [27]. In addition, A_2A_R density in lymphocytes also increased in patients with coronary artery disease [28] and chronic heart failure (CHF) [29]. This link between A_2A_R and PBMCs is now revealed for AF, although the specific mechanism underlying that remodelling of A_2A_R has not yet been determined. Moreover, the altered levels of adenosine and ADA activity in AF raises the question of whether the increased adenosine levels in AF are a cause or consequence of the disease. If peripheric adenosine augmentation triggers AF, then it might be cogitated if a reduction in ADA activity is behind AF pathophysiology. Interestingly, in CHF a reduction in ADA and ADA2 activity has been reported [30], thus, attenuating the pathologic consequences of CHF as putative increased levels of adenosine would compensate the cardiodepressant effects of increased TNFα in CHF patients [30]. Yet, here a lowering of ADA activity might be expected in patients at risk of AF, i.e., dSR, but this is not observed (although ADA levels appear to be slightly reduced and adenosine levels slightly increase in dSR). Alternatively, it can be speculated that the reduction in ADA activity is one step further in the different events that ends in an AF or can be just another risk-factor for the AF, as well as high blood pressure, age, or obesity. Conversely, the higher adenosine levels in plasma from AF patients might reflect excessive energy consumption in atria with myocytes beating at abnormally high frequencies. The higher adenosine levels can be sufficiently elevated to preclude a correct ADA catabolism, and thus, to induce an abnormal adenosinergic signalling, which could trigger the observed PBMC and myocytic A_2A_Rs remodelling. 

Interestingly, we found a correlation between adenosine levels and A_2A_R expression. Elevated adenosine levels and A_2A_R expression observed in patients with AF supports the notion that excessive A_2A_R activation contributes to a higher incidence of spontaneous calcium release-induced electrical activity and irregular beating reported in atrial myocytes from patients with AF [17,31], but what mechanisms are connecting adenosine levels and A_2A_R expression? On one hand, a possible explanation is a positive regulatory loop between adenosine and A_2A_R where adenosine is released to the environment due to excessive and abnormal beating and a consequence of the energy spent on it. This release would induce a positive turnover of A_2A_R expression, which could be necessary to keep the heart in a fast-beating condition. On the other hand, adenosine levels are probably a consequence of AF, which leave us the possibility that A_2A_R enhanced expression is the first step on the AF progression. This leads to abnormal heart beating due to a disturbance in calcium homeostasis and triggering higher adenosine levels. This would suggest a causative correlation, where the A_2A_R expression is altered prior to the adenosine content. In addition, there could be other, and even more complex, reasons behind the increased A_2A_R density and adenosine levels in AF, which might promote both simultaneously, without any apparently relationship between them. 

Overall, endogenous adenosine plays a key role on central cardiovascular regulation and it has profound modulatory effects through systemic mechanisms of autonomic regulation [32]. Thus, it has been suggested that adenosine may provide a connection between the local mechanisms of blood flow regulation and the systemic processes of autonomic cardiovascular modulation [32]. Hence, upon increased metabolic demand (exercise) and decreased energy supply (ischemia) interstitial myocardial levels of adenosine rise and activate sensory afferent (metaboreceptors) triggering a sympathetic systemic vasoconstriction, thus increasing blood pressure, and improving perfusion pressure. Indeed, while this systemic vasoconstriction would be harmful to the ischemic organ, a local adenosine A_1_R-dependent anti-adrenergic vasodilatory effects will protect the myocardium, whereas still benefited by the improved perfusion pressure. Thus, further research is needed to reveal the mechanism underlying the precise interplay between A_2A_R and A_1_R in AF. 

## 4. Materials and Methods

### 4.1. Reagents

The following reagents were used: adenosine deaminase (ADA, Roche Diagnostics, Mannheim, Germany), MRS7396 [21,33], ZM241385 (Tocris, Bristol, UK), Hoechst 33342 (Thermo Fisher Scientific, Rockford, IL, USA). The antibodies used were: mouse anti-A_2A_R (sc-32261, Santa Cruz Biotechnology Inc., Dallas, TX, USA), rabbit anti-α-actinin (sc-17829; Santa Cruz Biotechnology Inc.), horseradish peroxidase (HRP)-conjugated rabbit anti-mouse IgG (Pierce Biotechnology, Rockford, IL, USA) and HRP-conjugated goat anti-rabbit IgG (Pierce Biotechnology, Rockford, IL, USA).

### 4.2. Human Samples (Subject Demographics)

Human whole blood Samples (4 mL) from 28 non-dilated sinus rhythm (ndSR), 24 dilated sinusal rhythm (dSR) and 27 atrial fibrillation (AF) patients (Hospital Sant Pau Dos de Maig, Barcelona) (Table 1) were collected into lithium heparin tubes. In addition, atrial heart tissue from 11 ndSR and 12 AF patients undergoing cardiac surgery and stored immediately at −80 °C. Key clinical and echocardiographic data, as well as pharmacological treatments of these patients, included in the statistical analysis are summarized in Table 1. Patients undergoing mitral valve replacement or repair were not included in this study to avoid potentially confounding the effects of mitral valve disease. Each patient gave written consent for a sample of the right atrial appendix to be given that would have otherwise been discarded during the surgical intervention. The study was approved by the Ethics Committee at Hospital de la Santa Creu i Sant Pau, Barcelona, Spain, and the investigation conforms to the principles outlined in the Declaration of Helsinki.

### 4.3. Membrane Preparation

Right atrial samples were pulverized in liquid nitrogen congelation and sonicated (Brandson Sonifier 250, ICN Hubber S.A.) in 500 μL of ice-cold 10 mM Tris HCl, pH 7.4 buffer containing a protease inhibitor cocktail (Roche Molecular Systems, Belmont, CA, USA). The sonicated tissue was further homogenized using a Polytron (VDI 12, VWR, Barcelona, Spain) for three periods of 10 s each. The homogenate was centrifuged at 12,000 × *g* at 4 °C for 30 min. The membranes were dispersed in 50 mM Tris HCl (pH 7.4) and 10 mM MgCl_2_, washed, and resuspended in the same medium as described previously [34]. Protein concentration was determined using the BCA protein assay kit (Thermo Fisher Scientific Inc., Rockford, IL, USA) and 10 μg of protein was used for immunoblotting. 

### 4.4. Gel Electrophoresis and Immunoblotting

Sodium dodecyl sulphate polyacrylamide gel electrophoresis (SDS/PAGE) was performed using 10% polyacrylamide gels. Proteins were transferred to Hybond^®^-LFP polyvinylidene difluoride (PVDF) membranes (GE Healthcare, Chicago, IL, USA) using the Trans-Blot^®^TurboTM transfer system (Bio-Rad, Hercules, CA, USA) at 200 mA/membrane for 30 min. PVDF membranes were blocked with 5% (wt/vol) dry non-fat milk in phosphate-buffered saline (PBS; 8.07 mM Na_2_HPO_4_, 1.47 mM KH_2_PO_4_, 137 mM NaCl, 0.27 mM KCl, pH 7.2) containing 0.05% Tween-20 (PBS-T) during 1h at 20 °C before being immunoblotted using mouse anti-A_2A_R (0.5 µg/mL) and rabbit anti-α-actinin (0.5 µg/mL) antibodies in blocking solution overnight at 4 °C. PVDF membranes were washed with PBS-T three times (5 min each) before incubation with either a HRP-conjugated rabbit anti-mouse IgG (1/10,000) or HRP-conjugated goat anti-rabbit IgG (1/30,000) in blocking solution at 20 °C during 2 h. After washing the PVDF membranes with PBS-T three times (5 min each), the immunoreactive bands were developed using a chemiluminescent detection kit (Thermo Fisher Scientific) and detected with an Amersham Imager 600 (GE Healthcare Europe, Barcelona, Spain).

### 4.5. RT-qPCR

Peripheral blood mononuclear cells (PBMCs) were isolated from blood samples by density centrifugation using Lymphoprep (Lymphoprep, Palex Medical, Madrid, Spain). PBMCs were stored at −80 °C in Triazol until use. Cell RNA was purified using RNeasy Plus Universal Mini Kit (Qiagen, Hilden, Germany). Heart tissue was pulverized upon liquid nitrogen congelation and sonicated in 100 µL of Tris (pH 7.4) buffer and stored with 1ml of TriPure reagent (TriPure isolation Reagent, Roche) at −80 °C. Subsequently, the RNA was separated from DNA/proteins using TriPure (Sigma-Aldrich, St. Louis, MO, USA) and following manufacturer’s instructions. RNA yield was determined using a NanoDrop^®^ ND-1000 UV-Vis spectrophotometer (Thermo Fisher Scientific, Rockford, IL, USA). Reverse transcription was performed using 0.5–1 µg of total RNA obtained from each sample using StaRT Reverse Transcription Kit (AnyGenes, Paris, France) according to the manufacturer’s instructions. To quantify A_2_AR mRNA expression, we used TaqMan single tube assay probe (ADORA2A Hs00169123_m1, Thermo Fisher Scientific, Rockford, IL, USA). The TBP (TATA-box binding protein) probe (TBP Hs00427620, Thermo Fisher Scientific, Rockford, IL, USA) was used to normalize mRNA expression from both PBMC’s and heart samples. Reaction conditions were done following manufacturer’s instructions in a 384-well plate (MicroAmp Optical 384-well plate; Applied Biosystems, Foster City, CA, USA). The reaction was carried out in a total volume of 10 µL per reaction. The reaction mix included 5 µL TaqMAn Fast Advanced Master Mix, 0.5 µL TaqMan Assay Probe, 3.5 µL of Nuclease Free Water and 1µL of cDNA template (100 ng cDNA total amount) and were done in 7900HT Advanced Real-Time PCR instrument (Applied Biosystems). Amplification protocol started with 95 °C for 10 min, 42 cycles of 95 °C (10 sec) and 60 °C (30 sec), a final cycle of 95 °C (10 sec), and finishing with a 60 °C (30 sec) step. The results are presented relative to those for the housekeeping gene TBP using the ΔCq method. Data were analysed with Expression Suit Software v1.2 (Thermo Fisher Scientific, Rockford, IL, USA).

### 4.6. Sample Preparation and HPLC-MS/MS Determination of Ribonucleosides

Quantitative analysis of ribonucleosides (i.e., adenosine) was carried out as previously described [29,35,36]. In brief, we used liquid chromatography technique coupled to tandem mass spectrometry (LC-MS/MS) using an Agilent 1290 Infinity UHPLC chromatograph (Santa Clara, CA, USA) coupled to a 6500 QTRAP mass spectrometer (ABSciex, Framingham, MA, USA) equipped with Ion Drive Turbo V ion source operating in positive ion mode. The column used was a Discovery HS F5-3 150 × 2.1 mm 3µm (Supelco, Bellefonte, CA, USA) at 40 °C; autosampler temperature, 4 °C; injection volume, 10µL; flow rate, 0.6 mL min^−1^. Mobile phase was A) Ultrapure water with 0.1% HCOOH and B) Acetonitrile with 0.1 %HCOOH. The gradient program was as follows (t, %B): (0, 1), (0.2, 1), (3, 20), (3.5, 95), (6, 95), (6.5, 1), (10, 1). Mass spectrometry detection was performed by using the multiple reaction monitoring (MRM) mode using the following parameters: ion spray voltage, +5500 V; source temperature, 600 °C; curtain gas, 20 psi; ion source gas 1 and gas 2, 50, and 20 respectively; collision-activated dissociation gas, High; entrance potential, (+/-)10 V. The MRM transitions for adenosine were 268.3/136.1 (Declustering potential DP 65V and collision energy CE25V) for quantitative purposes and 268.3/119.1 (DP 60V, CE 25V) for confirmation purposes dA-d_3_ was used as internal standard with a transition 230.9/115 (DP 20V, CE 15V). The calibration curve was constructed with adenosine standard solutions between 0.03–8 ng/mL diluted in water TCA30%. Linear regression was adjusted (1/x or 1/x^2^) to have accuracies between 80-120% for all the adenosine standards. Analyst 1.6.2 Software was used for data acquisition and MultiQuant 3.0.1 for data processing both from ABSciex (Framingham, MA, USA).

### 4.7. Adenosine Deaminase Activity Determination

Plasma from each patient’s blood was isolated by density centrifugation using Lymphoprep (Lymphoprep, Palex) and stored at −80 °C until use. ADA was determined using a commercial colorimetric ADA assay kit (Dyazime Laboratories Inc., Poway, CA, USA) and following the manufacture’s indications. Adding the reagents of the kit you produce a colorimetric reaction based on quinone dye. Therefore, ADA activity in each plasma sample (5 µL) was determined in triplicate by measuring the absorbance at 550 nm using a POLARstar Omega plate reader (BMG Labtech, Ortenberg, Germany).

### 4.8. Cell Culture

Human embryonic kidney (HEK)-293T stable cell line permanently expressing the A_2A_R^SNAP^ construct [37] were maintained in Dulbecco’s modified Eagle’s medium (DMEM) (Sigma-Aldrich) supplemented with 1 mM sodium pyruvate, 2 mM L-glutamine, 100 U/mL streptomycin, 100 mg/mL penicillin and 5% (*v/v*) fetal bovine plasma at 37 °C and 5% CO_2_.

### 4.9. Flow Cytometry and Confocal Imaging

A_2_AR^SNAP^ expressing HEK-293T cells and human PBMCs (4 × 10^5^ cells) were incubated with 50 nM MRS7396 in presence or absence of 100 nM ZM241385 (Tocris Biosciences, Bristol, UK) in 300 µL of complete DMEM, or RPMI, respectively, under constant rotation (25 rpm, WiseMix Rotator, Witeg Labortechnik, Wertheim, Germany) during 16 h at 4 °C. Subsequently, cells were centrifuged at 500× *g* for 5 min at 4 °C and rinsed once with ice-cold Hank’s balanced salt solution (HBSS; Sigma-Aldrich, St. Louis, MO, USA) and immediately used for flow cytometric measurements or confocal imaging.

Flow cytometry measurements were conducted as previously described [38]. In brief, for single MRS7396 binding detection a total of 10^4^ cells (events) were analysed using a BD FACSCanto II flow cytometer (Becton Dickinson, Heidelberg, Germany) with excitation at 488 nm and emission at 647 nm. Samples were maintained in the dark and at 4 °C during the analysis to avoid photobleaching and ligand dissociation. PBMCc were selected by size and shape, excluding the death cells, triplets and doublets, while the mean fluorescence intensity per event was obtained in the FL-1 channel in log mode. Data were collected using FacsDiva Software v6 1.3 (Becton Dickinson, Ashland, OR, USA) and the analysis was made using the FlowJo v10.1 software (Becton Dickinson, Ashland, OR, USA).

For confocal imaging cells were seeded with DMEM or RPMI medium containing 1 μg/mL of Hoechst 33342 (Thermo Fisher Scientific) in an 8-well treated plate (Ibitreat 15µ-Slide 8 well Plate, Ibidi, Gräfelfing, Germany) for 20 min at 4 °C. Confocal images were captured using ZEISS microscope (ZEISS LSM 880 Confocal Laser Scanning Microscope, Zeiss, Oberkochen, Germany). Analysis was done with ZEN software (Zen 2 blue edition, Zeiss). 

### 4.10. Data and Statistical Analysis

Data are represented as mean ± standard error of mean (SEM) with statistical significance set at *p* < 0.05. The number of samples/subjects (*n*) in each experimental condition is indicated in the corresponding figure legend. Outliers were assessed by the ROUT method [39], thus, subjects were excluded assuming a Q value of 1% in GraphPad Prism 9 (San Diego, CA, USA). Data normality was assessed by the Shapiro-Wilk normality test (*p* < 0.05). Comparisons among experimental groups was performed by Student’s t test or one-way analysis of variance (ANOVA), followed by Tukey’s multiple comparisons post-hoc test using GraphPad Prism 9, as indicated. The correlation coefficients (r) were calculated using the Pearson two-tailed correlation test.

## Figures and Tables

**Figure 1 ijms-22-03467-f001:**
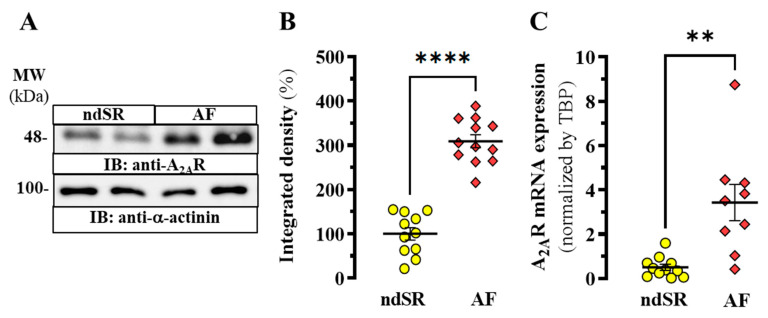
A_2A_R expression in human right atrium. (**A**) Representative immunoblot showing the expression of A_2A_R in right atrium from non-dilated sinus rhythm (ndSR) and atrial fibrillation (AF) patients; membranes from human atrium were analysed by SDS-PAGE (10 μg of protein/lane) and immunoblotted using goat anti-A_2A_R and rabbit anti-α-actinin antibodies (see Methods). (**B**) Relative quantification of A_2A_R density; the immunoblot protein bands corresponding to A_2A_R and α-actinin from ndSR (*n* = 11) and AF (*n* = 12) patients were quantified by densitometric scanning; values were normalized to the respective amount of α-actinin in each lane to correct for protein loading. (**C**) Relative expression of A_2A_R transcripts in human atrium. Mean ± SEM from ndSR (*n* = 11) and AF (*n* = 9) patients. **** *p* < 0.0001 and ** *p* < 0.01, Student *t* test.

**Figure 2 ijms-22-03467-f002:**
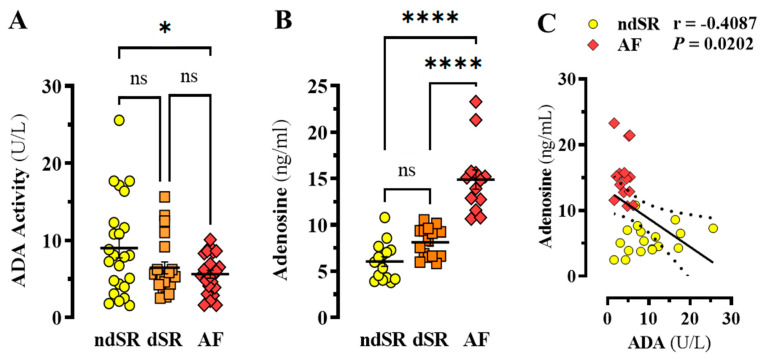
**ADA activity and adenosine content in human plasma.** (**A**) ADA activity in plasma from ndSR (*n* = 24), dSR (*n* = 22) and AF (*n* = 23) patients was determined using the Dyazime kit. Two outliers were removed from the dSR group by the ROUT method (*see* Materials and Methods). (**B**) Adenosine content in plasma from ndSR (*n* = 15), dSR (*n* = 15) and AF (*n* = 13) patients was determined by MS-HPLC. One outlier was removed from the AF group by the ROUT method (see Materials and Methods). (**C**) Correlation between ADA activity and adenosine content in ndSR (yellow, *n* = 18) and AF (red, *n* = 14) patients. The correlation coefficients (r) were calculated using the Pearson’s two-tailed correlation test. The results are expressed as mean ± SEM. * *p* < 0.05 and **** *p* < 0.0001, one-way ANOVA with Tukey’s post-hoc test.

**Figure 3 ijms-22-03467-f003:**
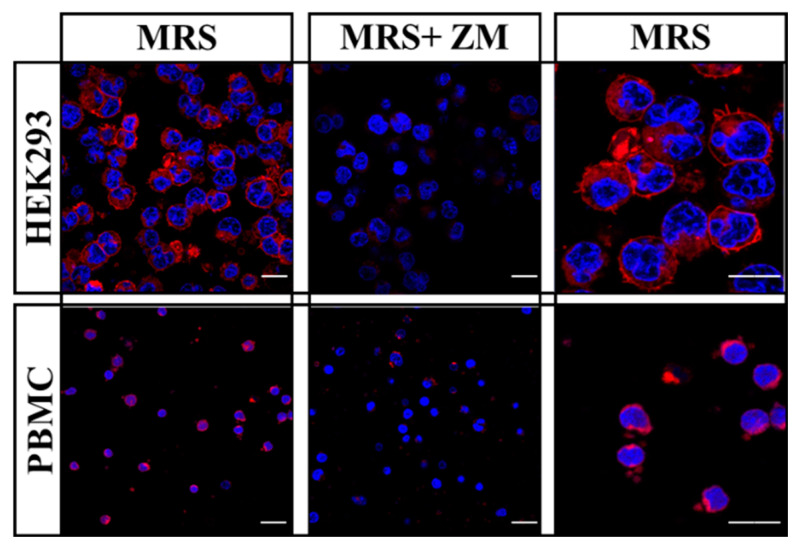
A_2A_R detection in living cells using a fluorescent ligand. HEK-293 cells permanently expressing A_2A_R and human peripheral blood mononuclear cells (PBMC) were incubated with 50 nM MRS7396 (MRS, red) in the absence or presence of 100 nM of ZM241385 (ZM). For live nuclear staining cells were incubated with 1µM Hoechst 33342 (blue). Scale bar: 20 μm.

**Figure 4 ijms-22-03467-f004:**
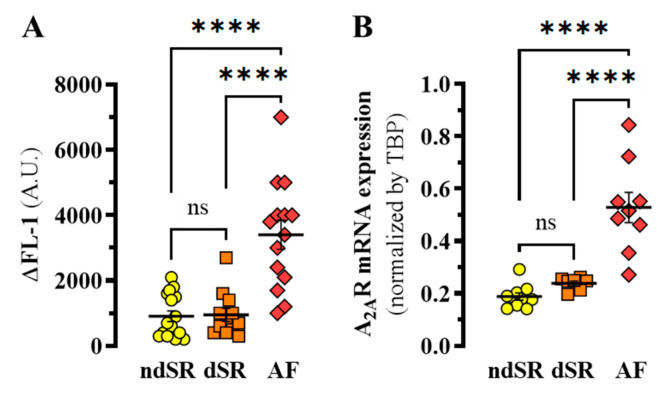
Detection of A_2A_R in human PBMCs. (**A**) The specific MRS7396 binding to PBMCs from ndSR (yellow, *n* = 16), dSR (orange, *n* = 12) and AF (red, *n* = 14)) patients was obtained by subtracting the nonspecific binding (i.e., 100 nM MRS7396 in the presence of 100 nM of ZM241385) and represented as ∆FL-1 (specific binding). (**B**) Relative expression of A_2A_R mRNA in PMBCs. Mean ± SEM of ndSR (*n* = 10), DSR (*n* = 7) and AF (*n* = 9) patients. **** *p* < 0.0001, one-way ANOVA with Tukey’s *post-hoc* test.

**Figure 5 ijms-22-03467-f005:**
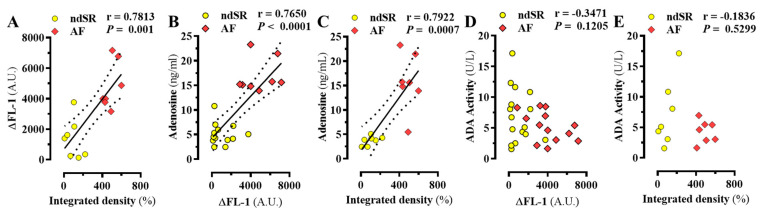
Correlation of atrial A_2A_R density and peripheric adenosinergic system. Correlation between the relative A_2A_R density in the right atrium determined by immunoblot (Integrated density, see Figure 1B) and the specific MRS7396 binding (∆FL-1) to PBMCc (**A**), adenosine content (**C**) and ADA activity (**E**) from ndSR (yellow) and AF (red) patients. Also, the correlation between the specific MRS7396 binding (∆FL-1) to PBMCc and the adenosine content (**B**) and ADA activity (**D**) from ndSR (yellow) and AF (red) patients was assessed. The correlation coefficient (r) was calculated using the Pearson (two-tailed correlation) test.

**Table 1 ijms-22-03467-t001:** Patients’ clinical information.

	ndSR	dSR	AF
Number of patients	28	24	27
Weight (mean±SD) (Kg)	83.18 ± 9.86	71.5 ± 10.5	75.5 ± 11.386
Height (mean±SD) (cm)	169.4 ± 9.8	160.8 ± 8.7	163.8 ± 8.9
Age (mean±SD)	63.1 ± 9.7	73.5 ± 8.4	72.9 ± 9.4
Sex (male/female)	23/5	17/7	14/13
Body Surface (mean ± SD) (m^2^)	1.93 ± 0.14	1.75 ± 0.15	1.81 ± 0.15
LA diameter index (mean ± SD)	1.99 ± 0.17	2.6 ± 0.2	2.65 ± 0.55
LV diameter index (mean ± SD)	2.7 ± 0.5	2.83 ± 0.54	2.82 ± 0.42
Tabaquism (y/n/ex)	9/12/7	3/18/3	6/18/3
Enolism (y/n/ex)	0/28/0	1/23/0	1/26/0
Hipertension	25	22	20
Diabetes	11	12	9
Dislipemia	15	18	15
Drugs			
IECAs	16	12	12
Beta-bloquers	12	9	16
β1	3	2	8
β2	0	0	1
α	0	0	0
RyR-inhibitors	0	0	0
ARA-II	5	4	2
Ca^2+^ antagonist	6	3	4
Sintrom	3	1	16
AAS	17	15	6
Statins	17	18	15

## Data Availability

Not Applicable.

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
