# Peer review of "Adenosine A2A Receptors Are Upregulated in Peripheral Blood Mononuclear Cells from Atrial Fibrillation Patients"

_ijms, 2021, doi:10.3390/ijms22073467_

Round 1

Reviewer 1 Report

The paper by Godoy-Marín et al. is well written and has scientific soundness. 

The present work focuses on the major role of the A2AARs in atrial fibrillation, reporting that the characterization of this receptor could be a useful tool to detect the disease status and the efficacy of the treatments. Overall, these results are very significant and interesting for readers and I think that this paper is worthy of publication  in IJMS in the present form.

Author Response

We thank the reviewer his/her kind comments on our manuscript.

Reviewer 2 Report

In the present study the authors aim to assess the status of the adenosinergic system in AF by analysing the expression of A2ARs both in right atrium biopsies and peripheral blood mononuclear cells (PBMCs) from AF patients. In addition, the adenosine levels and adenosine deaminase (ADA) activity in plasma from the same patients were evaluated.

They report an increased A2AR expression in right atrium from AF patients, increased levels of adenosine content and reduced ADA activity in plasma from AF patients as compared to patients in sinus rhtyhm. Concerning  A2AR expression, an increase was observed with a positive correlation (P=0.001) between A2AR expression in right atrium and PBMCs

With these results the authors highlight the importance of the A2AR in AF and suggest that the evaluation of this receptor in PBMCs may be potentially useful for monitoring disease severity and the efficacy of pharmacological treatments in AF patients.

From my point of view, the paper is delicately presented, with robust results. Probably the discussions are a little bit more speculative and provocative. 

I do not have major comments to add. I simply miss some mention in the discussion among the potential relationship among the autonomic nervous system and adenosin levels, be the latter a marker of the former. Subsequently, i think is important to highlight how many patients suffered CHF (they discuss it but its not mentioned in the methods) and AF type, separating adenosin levels / A2AR according to paroxysmal and non-paroxysmal (persistent; long-standing persistent and permanent), although the number of patients is low and would be a hypothesis i think it is important to mention. 

Author Response

We thank the reviewer for his/her comments. Indeed, the reviewer is right, and we should mention in the discussion the relationship between autonomic nervous system and adenosine levels. Thus, in the new version of the manuscript we further discussed this issue (“Overall, endogenous adenosine plays a key role on central cardiovascular regulation and it has profound modulatory effects through systemic mechanisms of autonomic regulation [32]. Thus, it has been suggested that adenosine may provide a connection between the local mechanisms of blood flow regulation and the systemic processes of autonomic cardiovascular modulation [32]. Hence, upon increased metabolic demand (exercise) and decreased energy supply (ischemia) interstitial myocardial levels of adenosine rise and activate sensory afferent (metaboreceptors) triggering a sympathetic systemic vasoconstriction, thus increasing blood pressure, and improving perfusion pressure. Indeed, while this systemic vasoconstriction would be harmful to the ischemic organ, a local adenosine A1R-dependent anti-adrenergic vasodilatory effects will protect the myocardium, whereas still benefited by the improved perfusion pressure. Thus, further research is needed to reveal the mechanism underlying the precise interplay between A2AR and A1R in AF.”).

In the discussion section we mentioned that in chronic heart failure (CHF) patients an increased density of A2AR in lymphocytes has been reported previously. Also, in CHF patients a reduction in ADA and ADA2 activity has been described. As stated in Table 1, we used biological material from ndSR (non-dilated sinus rhythm, n=28), dSR(dilated sinus rhythm, n=24) and AF (atrial fibrillation, n=27) patients. Thus, we do not work with CHF patients (see Table 1), thus we can’t provide any number of patients suffering CHF. Nevertheless, we agree with the reviewer that studies with a larger number of patients and also including CHF patients, would be interesting to be performed in the future.

Reviewer 3 Report

This paper present interesting results about the involvement of A2AR in atrial fibrillation. The paper is well written and interesting. I particularly appreciated the effort to measure ADA activity and adenosine concentration in blood. These results are rarely presented despite the importance of adenosine signaling in many inflammatory diseases.

Author Response

(The authors gave the same response as above.)
